# Influence of Laparoscopic Surgery on Cellular Immunity in Colorectal Cancer: A Systematic Review and Meta-Analysis

**DOI:** 10.3390/cancers15133381

**Published:** 2023-06-28

**Authors:** Annika Bohne, Elena Grundler, Helge Knüttel, Alois Fürst, Vinzenz Völkel

**Affiliations:** 1Fakultät für Medizin, Universität Regensburg, Universitätsstraße 31, 93053 Regensburg, Germany; 2Universitätsbibliothek Regensburg, Universität Regensburg, Universitätsstraße 31, 93053 Regensburg, Germany; 3Caritas Krankenhaus St. Josef Regensburg, Klinik für Allgemein-, Viszeral-, Thoraxchirurgie und Adipositasmedizin, Landshuter Str. 65, 93053 Regensburg, Germany; 4Tumorzentrum Regensburg—Zentrum für Qualitätssicherung und Versorgungsforschung der Universität Regensburg, Am BioPark 9, 93053 Regensburg, Germany

**Keywords:** laparoscopy, open surgery, surgical stress response, cellular immunity, natural killer cells, lymphocytes, leukocytes, colon cancer, rectal cancer

## Abstract

**Simple Summary:**

Colorectal cancer (CRC) is mainly treated by surgical resection. However, surgery induces a state of immunosuppression and leads to attenuation of the cellular immunity that is so vital for successful defense against infections and malignant cells. Because this immunosuppression depends on the extent of the surgical trauma, it is hypothesized that minimally invasive surgical approaches such as laparoscopy have beneficial effects for cellular immunity, leading to fewer infectious complications as well as lower rates of locoregional recurrence and distant metastasis. Better short- and long-term oncologic outcomes are reported by clinical trials, and immunologic mechanisms might substantially contribute to these observations. Herein, the authors systematically compare minimally invasive laparoscopic surgery with open surgery in terms of their respective influences on certain aspects of cellular immunity.

**Abstract:**

Colorectal cancer (CRC) is the third most common cancer worldwide. The main treatment options are laparoscopic (LS) and open surgery (OS), which might differ in their impact on the cellular immunity so indispensable for anti-infectious and antitumor defense. MEDLINE, Embase, Web of Science (SCI-EXPANDED), the Cochrane Library, Google Scholar, ClinicalTrials.gov, and ICTRP (WHO) were systematically searched for randomized controlled trials (RCTs) comparing cellular immunity in CRC patients of any stage between minimally invasive and open surgical resections. A random effects-weighted inverse variance meta-analysis was performed for cell counts of natural killer (NK) cells, white blood cells (WBCs), lymphocytes, CD4+ T cells, and the CD4+/CD8+ ratio. The RoB2 tool was used to assess the risk of bias. The meta-analysis was prospectively registered in PROSPERO (CRD42021264324). A total of 14 trials including 974 participants were assessed. The LS groups showed more favorable outcomes in eight trials, with lower inflammation and less immunosuppression as indicated by higher innate and adaptive cell counts, higher NK cell activity, and higher HLA-DR expression rates compared to OS, with only one study reporting lower WBCs after OS. The meta-analysis yielded significantly higher NK cell counts at postoperative day (POD)4 (weighted mean difference (WMD) 30.80 cells/µL [19.68; 41.92], *p <* 0.00001) and POD6–8 (WMD 45.08 cells/µL [35.95; 54.21], *p <* 0.00001). Although further research is required, LS is possibly associated with less suppression of cellular immunity and lower inflammation, indicating better preservation of cellular immunity.

## 1. Introduction 

Any surgical intervention comes with varying degrees of controlled tissue damage, inducing neuroendocrine activation, a hypercoagulable state, and a proinflammatory response [1,2,3,4]. The latter soon provokes a compensatory anti-inflammatory state, leading to the final stage of the surgical stress response being potentially severe attenuation of host cellular immunity [5,6]. Normally, cells of the innate and adaptive immune system possess the ability to recognize exogenic structures, allowing for targeted defense against infections. However, if cellular immunity is suppressed by the surgical intervention itself, there exists an elevated risk for infections, even predisposing to septic complications [7,8,9].

Moreover, immune cells such as natural killer (NK) cells and certain T lymphocytes exhibit cytotoxic capacities. They are able to recognize malignant cells and induce their apoptosis, hence playing a key role in antitumor immunity. Impairment of cell-mediated immunity therefore leads to a paradoxical effect of cancer surgery: intended as a curative method, surgical immunosuppression may awaken dormant cancer cells, thus facilitating local and distal tumor recurrence and enhancing metastasis formation [10,11,12,13]. Therefore, better preservation of cellular immunity has repeatedly been linked to longer disease-free survival, while immunosuppression comes with a higher incidence of local recurrence and distant tumor metastasis [14,15,16].

Nonetheless, surgical resection is the modern treatment of choice for nonmetastatic colorectal cancer (CRC) [17]. And rightly so, as only complete surgical resection can offer patients a curative outcome in most cases, outweighing the disadvantages of surgery. CRC is currently the third most common cancer worldwide, causing nearly 1 million deaths in 2020 [18], with an increasing incidence unfortunately still to be expected [19]. With such an epidemiologic burden, interest in surgical developments that help to achieve better short- and long-term outcomes is high. Much attention has thus been paid to minimally invasive techniques: whereas open surgery was the standard method for decades, much potential is now seen in laparoscopic and robotic surgery. Clinical trials comparing these surgical approaches have not disappointed, as studies could repeatedly and consistently show benefits for laparoscopically operated patients, including lower anastomotic leakage rates, faster return of bowel function, lower incidence of surgical wound infections, and a general decline in infectious complications, thus allowing for faster recovery with a shorter length of hospital stay and more favorable morbidity rates than seen after open surgery [20,21,22,23]. Moreover, after years of established implementation of laparoscopic surgery, evidence is mounting that laparoscopy holds oncologic benefits in terms of lower recurrence rates and longer disease-free survival [24,25,26,27]. 

The link between these observed benefits and the minimally invasive procedure is believed to lie in the less traumatic nature of laparoscopic methods, which thus represent a smaller stimulus for the host stress response and allow for better preservation of immunocompetence [28]. 

To test this hypothesis, comparisons of immunity parameters between the different surgical approaches are needed. Systematic reviews and meta-analyses [29,30,31,32] have addressed this topic but were confounded by only small sets of high-quality studies and included also benign diseases; furthermore, previous meta-analyses are limited to publications up until 2003, and a dire need for an up-to-date analysis therefore exists. As the immunologic impact of surgery still receives wide interest and the techniques available for CRC resections are subject to constant change—thus demanding a review of the current literature—the authors herein aim to provide an up-to-date and rigorous look at immunity in the context of different surgical approaches. This meta-analysis and systematic review focuses on cellular immunity; humoral aspects of postsurgical immunity are evaluated elsewhere. 

## 2. Materials and Methods

The methods applied in this meta-analysis and systematic review were prospectively registered at PROSPERO (CRD42021264324) and the PRISMA 2020 checklist was used for writing the report [33]. A draft search strategy was published in a public repository [34].

### 2.1. Search Strategy and Eligibility

#### 2.1.1. Eligibility

To be eligible, studies had to address the systemic postoperative cellular immune response in patients with confirmed CRC of any stage. Elective surgical tumor resection by minimally invasive techniques (laparoscopic or robotic surgery) had to have been compared with open techniques in the context of an RCT.

#### 2.1.2. Study Identification

Online bibliographic databases and trial registries, namely MEDLINE (via Ovid), Embase (via Ovid), the Cochrane Database of Systematic Reviews (CDSR) and the Cochrane Central Register of Controlled Trials (CENTRAL; via Cochrane Library, Wiley), the Science Citation Index Expanded (via Web of Science Core Collection), Google Scholar, ClinicalTrials.gov, and World Health Organization (WHO) ICTRP, were searched. This search was last executed on 10 December 2021. An initial search strategy was developed for MEDLINE and thereafter adapted to the other databases choosing appropriate search syntax and index terms. By using a broad range of synonyms and thesaurus terms, the search strategies aimed for high sensitivity. No limits were applied to date, language, or study type. The PICO framework was used to build the search strategy: population: colorectal cancer; intervention: laparoscopy; control: open surgery. Full reproducible search strategies and additional details of the searches as well as a PRISMA-S checklist [35] are available in a public repository [36,37]. The reference lists of systematic reviews relevant to this subject [29,30,32] and of the included studies were screened for further relevant records.

Deduplication according to the Bramer method [38] in Endnote (semiautomatic steps A–C) was followed by further deduplication using the Systematic Review Accelerator [39]. Deduplicated articles were screened for eligibility first by title and abstract and subsequently by full-text assessments using Rayyan [40] and only included if they were in accordance with the aforementioned PICO aspects and the study design. If full texts only insufficiently addressed these PICO aspects and the study design, they were excluded. If full texts were not available in English or German, they were excluded due to limited resources. The screening process was independently performed by two authors (EG, AB). Discrepancies were solved by discussion after the screening concluded. Reference lists of included articles and recent systematic reviews were screened for further relevant articles by one author (AB).

### 2.2. Data Collection

The “Data Collection Form—Intervention Review—RCTs only” (Cochrane Collaboration) [41] was adapted to this review’s question and used for data extraction. It included information on eligibility, publication details, characteristics of the study population (age, sex, cancer stage, numbers of participants converted from LS to OS, immunomodulatory medication), characteristics of study design (allocation concealment, method of randomization, number of participants included/randomized/analyzed), and characteristics of the surgical techniques (operating time, incision length, type of anesthesia, operative method/tumor location). If conversions were not reported, it was assumed that no conversions had been performed. Study authors were contacted once via email in case of required clarifications.

All measurements of pre- and postoperative parameters of cellular immunity up to 8 days after surgery were considered. Grouping of these reported sampling timepoints was applied according to the following prospectively defined timeframes: 0–2 h, 3–9 h, and 10–15 h after surgery, and postoperative day (POD)1, POD2, POD3, POD4, POD5, and POD6–8. The extracted outcome data included measuring method, units, effect size, measure of variance, number of participants per surgical group, and sampling timepoint. If data were merely presented graphically, WebPlotDigitizer [42] was used to extract information.

Data collection was performed independently by two authors (AB, EG). 

### 2.3. Data Analysis

A narrative synthesis was performed for all cellular parameters reported by at least two studies. 

Additionally, meta-analysis was performed for all parameters with sufficient data by at least two studies in the required data format mean value and standard deviation. 

Before the statistical synthesis, absolute numbers were converted to the most common unit. If data were presented as medians and/or interquartile range, means and standard deviations were acquired by implementing the estimation method of Wan et al. [43]. If both standard care and fast-track care were reported within the LS and/or OS group of one study, these subgroups were both included in the meta-analysis and pooled according to formulas implemented in Review Manager (RevMan) [44], whereas a default administration of immunomodulatory corticosteroids was a reason to exclude the subgroup in question.

The following exception applied: if reported numbers deviated from those given by other studies by at least a factor of 1000, the data were deemed unrealistic and excluded from meta-analysis if clarification from the study authors could not be obtained. This was decided upon to avoid distortion of weighting during meta-analysis.

Meta-analysis was then performed using RevMan. The weighted inverse variance approach (DerSimonian–Laird random effects method) was used to calculate the weighted mean difference (WMD) accompanied by 95% confidence intervals ([lower limit, upper limit]). The presence of heterogeneity was assessed using the Q-statistic, I², and τ². An I² of up to 30% was defined as low, 60% as moderate, 80% as substantial, and up to 100% as considerable heterogeneity. Due to the low power of tests for heterogeneity, a *p*-value <0.1 was defined as significant; otherwise, a two-sided *p*-value of <0.05 was considered significant.

Additionally, to test the robustness of data originating from the estimation method of Wan et al. [43], sensitivity analysis with exclusion of respective data was performed. 

Furthermore, graphical presentation of relative changes in cellular parameters was performed for those already being considered in meta-analysis to further highlight their postoperative developments compared to baseline measurements. For this purpose, the studies with the minimally and maximally reported change in means were identified and plotted, with the resulting corridor including all other possible mean changes seen in other studies. These relative changes in means were either directly extracted or calculated analogously to the meta-analysis’ methods. 

### 2.4. Methodologic Quality and Bias Assessment

The Cochrane Collaborations’ Risk of Bias (RoB)2 tool [45] was used to assess the RoB. Judgements were independently made by two authors (EG, AB), with discrepancies solved by discussion and consultation with the senior author (VV). Study-level RoB ratings comprised “low”, “some concerns”, or “high”. Because the analysis by intention to treat (ITT) is unsuitable for assessment of the effect of adhering to the intervention, such an analysis method was a reason to uprate the RoB. Publication bias was assessed by comparing the results reported with the outcomes and measurement timepoints planned prospectively to identify selective nonreporting. Moreover, tests for funnel plot asymmetry were planned for analyses including at least ten studies, which is in concordance with recommendations of the Cochrane Collaboration [46]. As all meta-analyses included less than ten studies, this test could not be performed. If concerns of publication bias were raised, the quality of the generated results of analyses including the concerned studies were rated down by one level using the GRADE approach and the GRADE pro GDT software [47]. GRADE was used to rate the quality of evidence and for communication of confidence in results obtained, with overall quality ratings of “high”, “moderate”, “low”, or “very low”.

## 3. Results

### 3.1. Study Selection

This systematic literature search yielded 13,714 records from databases and registers. The precise sources can be found in the PRISMA flow diagram (Figure 1) and the Appendix A [37]. Duplicates were removed, with 7678 records remaining for screening of the title and abstract. Full texts were sought for 124 reports, and of these, 16 reports were included. For four reports, full texts could not be retrieved. All other reasons for exclusion are shown in Figure 1. One further study was identified only by searching the reference lists of the included records. Of the overall 16 records included, there were two cases of two records being based on the same study population (judged by population characteristics, authors, and publication date). Thus, 14 studies were included in this review of cellular immunity.

### 3.2. Characteristics of Included Studies

A total of 974 patients were included in this review. No study reported significantly differing sociodemographic characteristics, significantly deviating baseline values of cellular parameters, or significantly differing numbers of included tumor stages between the groups. Overall, all Union for International Cancer (UICC) stages were represented in this systematic review. UICC stage 0 was included by one study [48] and present in eight patients in the open group, but no patients in the LS group. UICC stage I was present in 65 patients of the LS group and in 96 patients of the OS group. Overall, UICC stage II was the most common: 184 and 198 patients had this stage, in the LS and OS groups, respectively. UICC stage III was also more frequent with 159 patients in the LS group compared to 123 in the open group. Because 8 studies had already prospectively excluded UICC stage IV, only 18 patients with this stage were included in the LS group and 9 in the OS group. Hence, less advanced stages contributed mostly to the results of this systematic review. The study of Laforgia et al. [49] must be mentioned explicitly, as it mostly evaluated more advanced stages, especially in the LS group. Four studies did not provide specific numbers on the UICC tumor stages included, with one of these studies stating to have excluded stage IV. Information on tumor stages can be found in Table 1 and in more detail in Appendix A [36].

Conversions from laparoscopy to open surgery were performed in eight studies, three did not perform conversions, and three did not specify whether conversions were performed. Two studies were limited to colonic resections, and two to rectal resections, and the remaining nine studies had performed rectal as well as colonic resections. Table 1 shows an excerpt of the characteristics of the included studies, with detailed information among the Appendix A).

### 3.3. Results of Analyses

A tabular overview of the narrative synthesis can be found in a public repository. 

Generally, leukocytes—as an indicator of postsurgical inflammation—rose, starting 3–9 h after surgery in both surgical groups, whereas the absolute cell numbers of lymphocytes and certain subsets showed a decline, most prominently observed around or at POD1. Of all 14 studies included, 7 reported cellular outcomes favoring the laparoscopic approach for at least 1 measurement timepoint, whereas 1 study found favorable as well as less favorable outcomes in the LS group regarding different parameters. 

A meta-analysis and graphical evaluation were possible for the white blood cell (WBC) count, number of NK cells (NK cell count; CD3−, CD16+, and/or CD56+), ratio of CD4+ T-helper lymphocytes to CD8+ T lymphocytes, total number of lymphocytes, and CD4+ T-helper lymphocyte count, with the results presented below. Significant differences favoring LS were found for NK cell counts at POD4 and POD6–8.

Several studies were ineligible for graphical and/or quantitative analysis. Justifications can be found in Appendix B.

#### 3.3.1. White Blood Cell Count

The results of the studies regarding the WBC count were heterogenous: of 8 studies evaluating this parameter, Laforgia et al. [49] reported higher WBC after LS at 4 h after surgery, whereas Duque et al. [50] (at POD1) and Ordemann et al. [56] (at POD1, 2, 4) found significantly lower WBCs (Appendix A). As leukocytosis is commonly seen after surgery, a lower WBC count is favorable, as it indicates less inflammation [64].

Compared to the baseline measurements, the WBCs were observed to rise in both the LS and OS groups directly after surgery, with the most pronounced relative rise being reported at 3–9 h after surgery in the LS group (2.8-fold increase), whereas the maximal rise after OS was reported for POD1 to a lower extent (1.9-fold increase). Interestingly, this relative rise was far more pronounced in the LS group, but because these data originated from one study only [49], with other studies reporting far lower relative increases for POD1, the informative value is limited. During the 8-day observation period, the mean WBC count in the OS group returned to preoperative values in one study (Laforgia et al. [49]) but remained elevated in other groups and studies (see Figure 2).

For the quantitative analysis (Figure 3), the data of Kvarnström et al. [53,54] had to be excluded because the cell numbers reported deviated from reasonable values (e.g., POD1 in LS: median 8.3 × 10^9^ cells/mL). The meta-analysis did not yield significant results for POD1 (WMD −0.55 × 10^3^ cells/µL [−2.87,1.76], *p =* 0.64), POD4 (WMD −0.63 × 10^3^ cells/µL [−1.97,0.71], *p =* 0.35), or POD6–8 (WMD 0.24 × 10^3^ cells/µL [−0.63,1.11], *p =* 0.59), and heterogeneity was considerable at POD1 (I^2^ = 93%, *p*-value of chi^2^-test *p <* 0.00001) but low or moderate at all other timepoints (POD4: I^2^ = 9%, *p =* 0.29; POD7: I^2^ = 41%, *p =* 0.19). Our results therefore indicate that LS compared to OS probably leads to little to no difference in the WBC count.

#### 3.3.2. Natural Killer Cell Count and Lytic Activity

Regarding NK cells, both the postoperative development of NK cell counts as well as NK cell activity as judged by cytotoxicity to target cells were evaluated. Both domains are commonly suppressed after major surgery [65]; hence, their preservation is a favorable outcome.

Shi et al. [57] reported significantly higher NK cell counts at POD4 and POD6–8 as well as better-preserved lysis activity at POD4 after LS. The other four studies also evaluating NK cell count and two studies focusing on NK cell lytic activity did not come to this conclusion, reporting nonsignificant differences for all sampling timepoints, although all studies uniformly reported a decline in NK cell numbers as well as lytic function, the latter especially pronounced at POD1. 

Upon combining the studies’ data to evaluate the relative postoperative development of NK cell numbers (Figure 4), a slight initial rise was observed up to 10–15 h (LS 137%, OS 114%), followed by a drop in cell counts with the minima reached by both surgical groups at POD3 (48% LS, 55% OS). 

The quantitative analysis of NK cell counts (Figure 5) resulted in no significant difference for POD1 (WMD −12.58 cells/µL [−35.19, 10.03], *p =* 0.28). However, at POD4, the mean cell counts differed markedly by 30.8 cells/µL (WMD 30.8 cells/µL [19.68, 41.92], *p <* 0.00001), being higher after LS. At POD6–8, the LS group again showed higher NK cell counts by 45.08 cells/µL (WMD 45.08 cells/µL [35.95, 54.21], *p <* 0.00001). Heterogeneity was low in all performed analyses (POD1: I^2^ = 22%, *p =* 0.28; POD4: I^2^ = 0%, *p =* 0.55; POD7: I^2^ = 0%, *p =* 0.8). Summarizing these findings, LS results in little to no difference compared to OS at POD1 but likely in a large reduction in NK cell count suppression at POD4 and POD6–8, thus leading to higher cell numbers at these timepoints.

#### 3.3.3. Lymphocytes and Subsets

Evidence regarding the cell numbers of lymphocytes and several of their subsets (CD3+, CD4+, CD8+ T lymphocytes, and B lymphocytes) is summarized below. Lymphocytes are commonly suppressed post-surgery with lymphopenia being associated with adverse outcomes [66,67,68]. Therefore, higher lymphocyte cell counts and subsets are favorable.

The total lymphocyte count [50,55,61,62] and the numbers of CD3+ [55,60,63], CD4+ [55,60,61,62], and CD8+ T cells [55,60,61,62] were all evaluated by three studies. Wang et al. [60] found significantly higher numbers of CD3+ T cells (at POD1, 3, 5) and CD4+ T cells (at POD1, 3) after LS, and Leung et al. [55] also reported higher CD8+ T cells after LS at POD6-8, whereas there were no significant differences regarding the total lymphocyte count. The number of B lymphocytes was evaluated by two studies [55,61,62], but none reported differences between the surgical approaches.

All studies reported declining numbers of lymphocytes and their subsets after both surgical approaches. In the current comparison to baseline values, the total numbers of peripheral lymphocytes (Appendix A) also declined, reaching relative minimum values at POD1 (LS 53%, OS 57%). Thereafter, cell counts rose again but did not reach preoperative dimensions during the observation period of up to 8 days (LS 86%, OS 75%). Similar developments were seen in the graphical analysis of CD4+ T cells (Appendix A): studies reported an approximate halving of the cell count at POD1 (LS 46%, OS 47%) for both surgical groups, with cell counts rising after this timepoint but not reaching preoperative values (maximum LS at POD3 with 85%, maximum OS at POD5 with 88%).

The meta-analysis of three studies [50,55,61,62] evaluated the total lymphocyte count at POD1 (Appendix A) but did not yield significant results (WMD 44.05 cells/µL [−10.46, 98.56], *p =* 0.11). Heterogeneity was not an issue in this analysis (I^2^ = 0%, *p =* 0.46). Additionally, the meta-analysis of two studies evaluating CD4+ T cells [55,61,62] at POD1 (Appendix A) also did not show a significantly different mean difference at POD1 (WMD 76.16 cells/µL [-47.12, 199.44], *p =* 0.23). Heterogeneity was low (I^2^ = 0%, *p =* 0.67). These analyses therefore indicate that the surgical approach likely made little to no difference regarding the lymphocyte count and CD4+ T cell numbers at POD1. 

#### 3.3.4. CD4+/CD8+ T Lymphocyte Ratio

Summarized narratively, Laforgia et al. [49] found significantly differing CD4+/CD8+ T lymphocyte ratios at 3-9 h and Wang et al. [60] at POD1 and POD3, with higher ratios after LS. The remaining four studies [48,52,56,61,62] did not report significant findings. Because a low ratio is indicative of immunodeficiency, a higher CD4/8 ratio is preferable [69].

The studies’ postoperative developments in the CD4+/CD8+ ratio (Appendix A) did not provide a clear picture, fluctuating around the preoperative values in both groups. The lowest ratio was observed 0–2 h after surgery in the LS group (67%), whereas the highest ratio was also reported after LS at 3–9 h after surgery (129%). At POD6–8, the CD4+/CD8+ ratio remained elevated except for in the OS group of Laforgia et al. [49]. The meta-analysis including four studies [48,49,52,60] (Appendix A) resulted in no significant differences between the groups at POD1 (WMD 0.06 [−0.01, 0.13], *p =* 0.08), POD3 (WMD 0.05 [−0.01, 0.12], *p =* 0.12), or POD6-8 (WMD −0.06 [−0.85, 0.74], *p =* 0.89). Heterogeneity was low or moderate (POD1: I^2^ = 0%, *p =* 0.94; POD3: I^2^ = 0%, *p =* 0.33; POD7: I^2^ = 48%, *p =* 0.17). The results of this meta-analysis hint toward laparoscopy probably resulting in little to no difference regarding the CD4+/CD8+ ratio when compared to OS.

#### 3.3.5. Monocytes and HLA-DR II Expression

The number of peripheral monocytes was reported by two studies [58,61,62], whereas five studies [52,56,58,59,61,62] evaluated the expression of HLA-DR II on the surface of CD14+ monocytes. Monocyte counts did not exhibit significant differences after differing surgical approaches, while Veenhof et al. [58] and Ordemann et al. [56] reported higher HLA-DR II expression rates on CD14+ monocytes after LS at 2 h and POD4, respectively. Because lower expression of HLA-DR II inversely correlates with favorable outcomes [70,71], a higher expression level should be considered beneficial.

### 3.4. Methodologic Quality of Included Studies

The overall risk of bias (Table 1) was rated as low in five studies. Seven studies raised some concerns regarding bias: allocation concealment was not clear in the studies by Duque et al. [50] and Hasegawa et al. [51], whereas the studies by Leung et al. [55], Tang et al. [48], Veenhof et al. [58], and Veenhof et al. [59] were uprated in terms of RoB due to conversions being analyzed by intention to treat. A high risk of bias was seen in two further studies: the studies by Kvarnström et al. [53,54] did not report allocation concealment; furthermore, they reported slightly unbalanced blood transfusions (1:4 in LS:OS) across intervention groups and did not report measurements of WBCs at two timepoints, as originally mentioned in the methods section. A prospective protocol to further classify the importance of this discrepancy was not available. This discrepancy also raised concerns of possible publication bias (due to selective nonreporting) regarding the meta-analysis results of the WBCs. As these studies results were only included in the sensitivity analyses, the main result of the meta-analysis at POD1 was not affected. The study by Laforgia et al. [49] was uprated for the RoB due to unknown allocation concealment. Additionally, concerning the WBC outcome, the cell numbers given deviated between text, graph, and table, potentially indicating repeated measurements. Therefore, the WBC outcomes of Laforgia et al. [49] were associated with high concerns of bias. Regarding the analysis, the numbers shown in the table were used, because numerical data were preferred over graphical data. Detailed information on the risk of bias judgements can be found in Appendix A [36]. 

When GRADEing the certainty of the evidence produced by the present meta-analyses, the effect size calculated for the NK cell count at POD1 was rated to be of high certainty. All other outcomes, except for the WBC at POD1, were classified as “moderate.” Downgrading by one level was justified because the optimal information size could not be reached due to small sample sizes. The WBC at POD1 was rated to be of low quality due to the relevant risk of bias in the WBC measurements of Laforgia et al. [49], and the presence of heterogeneity causing inconsistency.

## 4. Discussion

The current study investigated the impact of the different degrees of surgical trauma seen in laparoscopic and open surgery on the cellular immunity of CRC patients. For this purpose, the 11 parameters most reported by study authors were evaluated, and the meta-analysis found significantly better-preserved NK cell numbers and a general tendency toward lower markers of inflammation and better immunocompetence after LS. 

Postoperative recurrence and the occurrence of complications are closely related to the host’s inflammatory and immune state [8,72,73]. In both surgical groups across the included studies, postoperatively rising systemic WBC counts were observed, whereas lymphocyte numbers and subsets, HLA-DR II expression on monocytes, NK cell count, and NK cell activity generally drop. Hence, a shift in peripheral cell type proportions is seen in the postoperative course, as already described in other studies [68,74]. This indicates the presence of inflammation as well as postsurgical immunosuppression and is in accordance with the findings of other studies [6,10,15,64,67,75].

NK cells are crucial components of the antitumor innate immune response, with NK cell surveillance being especially effective at detecting and destroying nascent tumor cells [76]. The relevance lies particularly in the ability of NK cells to induce apoptosis in the absence of immunologic priming, making them the fast and first-line defense against malignant cells [77,78]. This leads to NK cell dysfunction being the “prime suspect in the case of paradoxical postoperative metastases,” as put by Market and colleagues [65], leading to increased rates of cancer recurrence and death, while a depletion in NK cell count was found to be an independent negative prognostic indicator for survival in colon cancer patients [79]. In the current meta-analysis, significantly higher counts of circulating NK cells after LS were found at POD4 and POD7, while Shi et al., in their quite recent and large-scale study, observed higher lytic abilities at POD4 after LS.

Generally, leukocytosis is seen as a physiologic reaction to trauma, indicating surgery-induced inflammation [80,81,82]. However, especially prolonged elevations of the WBC count come with a worse prognosis regarding disease-free and overall survival [82,83]; thus, less pronounced leukocytosis is a desirable postoperative outcome. In the present meta-analysis, WBC counts rose in both surgical groups but differed by a statistically nonsignificant degree, although results at the study level might indicate lower WBC counts after LS.

Monocytes normally facilitate antitumor immunity by presenting antigens to T-helper cells, mediating the antitumor abilities of cytotoxic T cells [84]. While monocyte counts did not differ between groups in the studies in this review, a significantly lower expression of HLA-DR II was seen after OS in two studies. However, low expression of HLA-DR II on CD14+ monocytes causes a conversion toward an anti-inflammatory phenotype, making the host more susceptible to infections and malignant processes. Low expression rates were therefore found to be an indicator of immunoparalysis during sepsis and are associated with increased mortality [70,71,85]. 

Besides innate immune cells, immunity relies heavily on the cells of the adaptive immune system. Although these cells require priming, and thus take longer to act as effector cells, they work in a much more targeted manner [86]. CD4+ T-helper cells aid other cells in their cytotoxic activity, activating NK cells and CD8+ T lymphocytes via cytokine release, with both cell types explicitly targeting cells degenerated due to viral infections or malignant processes [87,88]. T cell priming is a requirement for successful antitumor immunity [86], and diminished proportions of tumor-infiltrating CD8+ T cells are inversely correlated with tumor size [72]. However, surgery is known to induce apoptosis in T cells, leading to a reduction in cell numbers and hence attenuation of their important anti-infectious and antitumor capabilities [6,89]. In addition, the ratio of CD4+ to CD8+ T lymphocytes can be used as an indicator of immunocompetence, with a reduced ratio hinting at impairment of cellular immunity [90] and thus associated with higher risks of infectious complications. Altogether, better survival of CRC patients with higher CD3+, CD4+, and CD8+ lymphocyte subsets was reported by Milašienė et al. [91].

At the study level, higher counts of these lymphocyte subsets were repeatedly reported after LS but never after OS. In the present meta-analyses of circulating CD4+ T lymphocytes and total lymphocytes on the first postoperative day, the mean differences show tendencies toward higher cell counts after LS, albeit not reaching levels of significance. Similar results were seen for the CD4+/CD8+ ratio, which exhibited tendencies toward a higher and thus more favorable ratio after LS at POD1 and POD3, again without reaching significance.

In their preceding meta-analysis, Liu et al. [92] found significantly higher CD8+ T cells after laparoscopy at POD1–3 but no significant differences for the CD4+/CD8+ ratio and the CD4+ T cell number, which was therefore coherent with the results of the current analysis.

### Strengths and Limitations

With the inclusion of 14 studies and the data of 974 patients, this review is the most in-depth and up-to-date analysis on this subject. Due to the restriction to RCTs and the high quality of included studies, the authors’ confidence in the estimates of effect generated by the current meta-analyses is mostly moderate, in one case even high. Furthermore, the search strategy applied is another strength of this work, as its thorough approach allows for a reduction in publication bias [93] and higher confidence in the exhaustive identification of all relevant literature.

To ensure the greatest possible consideration of data, estimation methods were used to transform some of the studies’ summary measures. Despite the estimation methods used, the robustness of all the results of the meta-analyses was evident in the sensitivity analyses (Appendix A) applied.

Methodologically, this review was faced with a scarcity of corresponding published protocols for the included studies, thus limiting certain risk of bias judgements, such as evaluation of publication bias. Testing for funnel plot asymmetry was not possible, limiting the evaluation of publication bias. Moreover, although the work by Zhao et al. [94] is known to be of potential relevance to this review, it could not be included due to its full-text publication language.

The main restriction of this work lies in the limited amount of data included. Although it was possible to rely on a larger set of studies compared to previous works [29,30], several meta-analyses still did not reach the optimal information size. This was mostly due to the different choice of measurement timepoints of the study authors, rendering data pooling complicated. Furthermore, this review was limited to the postoperative timeframe up to 8 days, so relevant changes thereafter may have been missed. Regarding the included studies, only Hewitt et al. [52] measured parameters beyond postsurgical day 8. More investigation of the long-term influence on immunity might be relevant, as research indicates the importance of immunity far beyond the initial surgical insult [10,95,96,97]. This review includes all tumor stages, in particular UICC stages II and III. Thus, clarification of the extent to which different tumor stages influence the cellular immune response after surgery is a matter for further research. Moreover, future research should focus on evaluating parameters which are being used in everyday clinical practice (like WBC) allowing for real-world study settings, and for which there is sufficient evidence of their prognostic relevance (like NKC). Methodologically, the publication of prospective protocols and analysis plans would be highly desirable.

## 5. Conclusions

Upon comparison of absolute cell counts between different surgical approaches, the present analyses indicated better preservation of NK cells after laparoscopic surgery, as well as clear noninferiority regarding other parameters of cellular immunity and inflammation in comparison to open surgery. Thus, laparoscopy is a feasible option from an immunologic standpoint. With regard to the important antitumor domain of NK cells, laparoscopy might even allow for better preservation of the host antitumor defense so urgently needed for sustainable curative CRC surgery.

## Figures and Tables

**Figure 1 cancers-15-03381-f001:**
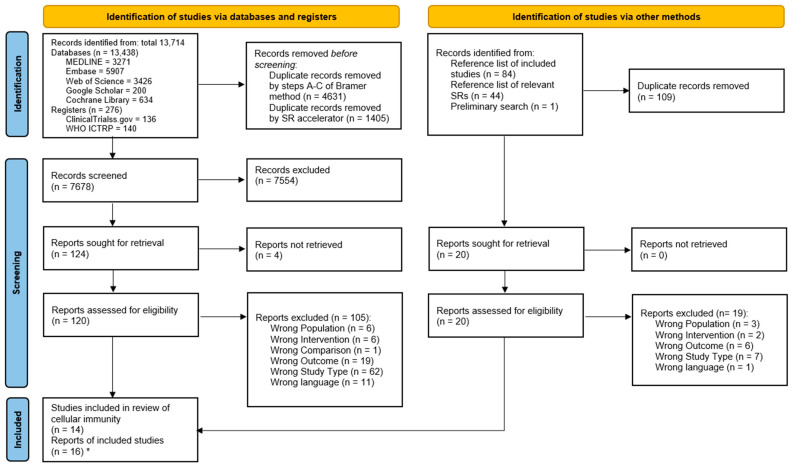
PRISMA flow diagram; * there were two cases of two records being based on the same study population; from Page MJ, McKenzie JE, Bossuyt PM, Boutron I, Hoffmann TC, Mulrow CD et al. The PRISMA 2020 statement: an updated guideline for reporting systematic reviews. BMJ 2021, 372, n71.

**Figure 2 cancers-15-03381-f002:**
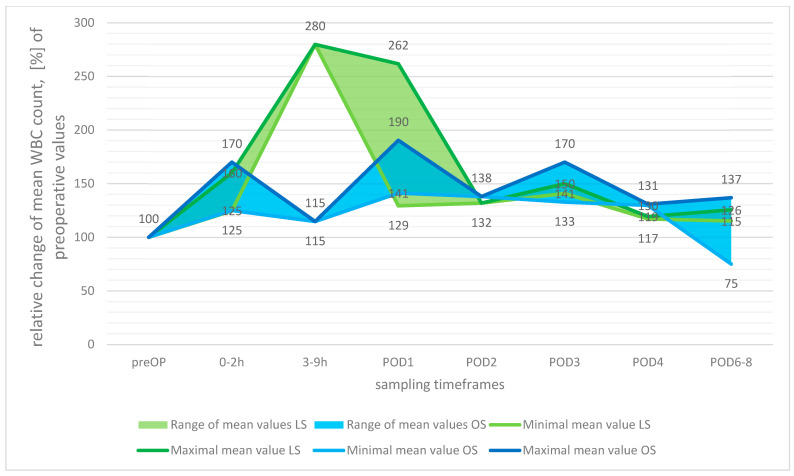
Postoperative development of white blood cell count (WBC), relative change in means with preoperative values set at 100%.

**Figure 3 cancers-15-03381-f003:**
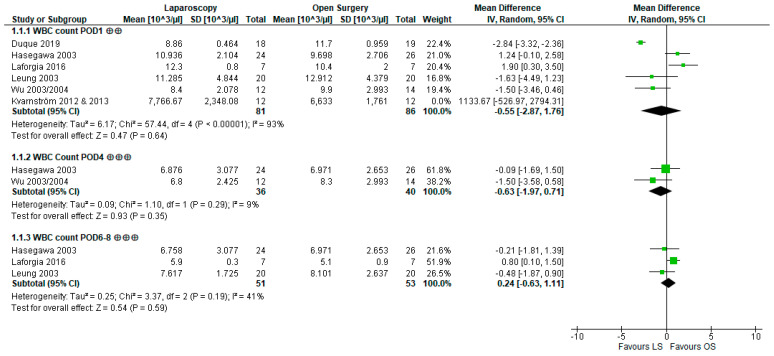
Forest plot of white blood cell (WBC) count at postoperative day (POD)1, POD4, POD6–8; ⨁⨁ indicates low, ⨁⨁⨁ indicates moderate confidence in estimates of effect; bold numbers indicate the pooled number of participants, pooled mean difference and the pooled 95% confidence interval with the mean difference of each study being indicated by a green square; data from Kvarnström et al. are depicted but not included in the analysis [49,50,51,53,54,55,61,62].

**Figure 4 cancers-15-03381-f004:**
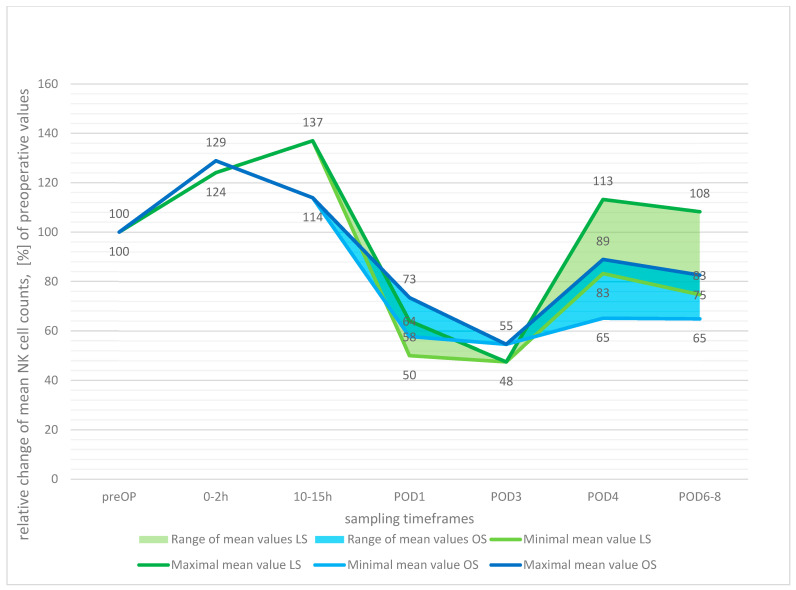
Postoperative development of natural killer (NK) cell count, relative change in means with preoperative values set at 100%.

**Figure 5 cancers-15-03381-f005:**
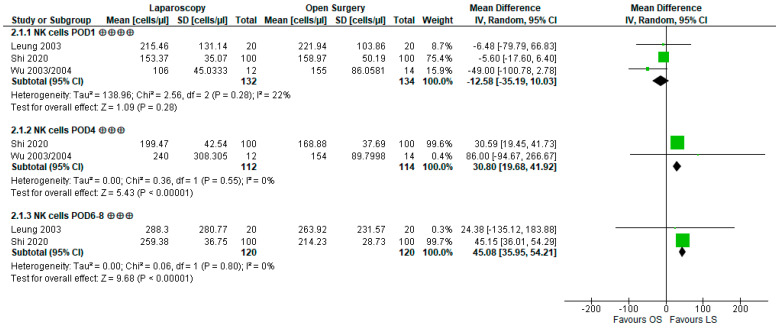
Forest plot of natural killer (NK) cell count at postoperative day (POD)1, POD4, POD6–8; ⨁⨁⨁ indicates moderate, ⨁⨁⨁⨁ indicates high confidence in estimates of effect; bold numbers indicate the pooled number of participants, pooled mean difference and the pooled 95% confidence interval with the mean difference of each study being indicated by a green square [55,57,61,62].

**Table 1 cancers-15-03381-t001:** Characteristics of included studies (for more detailed characteristics, see Appendix A).

First Author	Year of Publication	*n*	LS	OS	Inclusion of Stage IV	Colonic and/or Rectal Resections	Risk of Bias	Outcomes Evaluated
Duque [50]	2019	37	18	19	n.a.	Colorectal	Some concerns	WBC, Lym
Hasegawa [51]	2003	50	24	26	No	Colorectal	Some concerns	WBC, NKC Akt
Hewitt [52]	1998	16	8	8	No	Colorectal	Some concerns	CD4/8, HLA-DR II
Kvarnström [53,54]	2012/2013	24	12	12	n.a.	Rectal	High	WBC
Laforgia [49]	2016	14	7	7	Yes	Colorectal	High	CD4/8, WBC
Leung [55]	2003	40	20	20	No	Colorectal	Some concerns	NKC, NKC Akt, WBC, Lym, CD3+, CD4+, CD8+, BC
Ordemann [56]	2001	40	20	20	No	Colorectal	Low	CD4/8, WBC, HLA-DR II
Shi [57]	2020	200	100	100	No	Colorectal	Low	NKC, NKC Act
Tang [48]	2001	161	80	81	Yes	Colorectal	Some concerns	CD4/8, NKC
Veenhof [58]	2011	40	22	18	Yes	Rectal	Some concerns	WBC, HLA-DR II, Mono
Veenhof [59]	2012	79	42	37	n.a.	Colonic	Some concerns	HLA-DR II
Wang [60]	2012	163	80	83	No	Colonic	Low	CD4/8, CD3+, CD4+, CD8+
Wu [61,62]	2003/2004	26	12	14	No	Colonic	Low	CD4/8, NKC, WBC, Lym, CD4+, CD8+, HLA-DR II, BC, Mono
Xu [63]	2015	84	43	41	No	Colorectal	Low	NKC, CD3+

n.a. = information not available, *n* = total study population, LS = laparoscopically operated patients, OS = open surgery-operated patients, WBC = white blood cell count, Lym = lymphocyte count, NKC = natural killer cell count, NKC Act = natural killer cell activity, CD4/8 = CD4+/CD8+ ratio of T lymphocytes, CD3+ = cell count of CD3+ T lymphocytes, CD4+ = cell count of CD4+ T lymphocytes, CD8+ = cell count of CD8+ T lymphocytes, HLA-DR II = HLA-DR II expression on CD14+ monocytes, BC = B lymphocyte count, Mono = monocyte count.

## Data Availability

Data supporting the article were deposited in a publicly accessible repository (DOI 10.5283/epub.54402 & DOI 10.5283/epub.54403), references [36,37]. This includes the PRISMA 2020 checklist, PRISMA-S checklist, full reproducible search strategies, accession numbers of records found in searches, detailed characteristics of included studies table, tables of sensitivity analyses conducted, detailed risk of bias assessments, data collection forms of included studies, as well as the original graphs and forest plots.

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
