# Peer review of "Influence of Laparoscopic Surgery on Cellular Immunity in Colorectal Cancer: A Systematic Review and Meta-Analysis"

_cancers, 2023, doi:10.3390/cancers15133381_

Round 1

Reviewer 1 Report

The main question is the immunological status after mini invasive and open surgery.

The topic is relevant in the field. It adds a metanalisys and a personal study to the subject area compared with other published material.

I believe that the control study is ok; just like they say, moro case could be useful.

The conclusions are consistent with the evidence and arguments presented and they address the main question posed.

The references are appropriate.

In the paragraph  3.3.3, 3.3.4 and 3.3.5 are reported two, 3 and  3 references  while they are 3, 4 and 3; they associated some refernces (FE the voice 60 and 61 - of the some author, but they are two different voices.

Author Response

Response: We thank this reviewer for reviewing our manuscript and for contributing valuable comments on the issue of references. We now consistently quote the included studies and therefore added references in the sections 3.3.3 to 3.3.5:

Section 3.3.3: "Additionally, meta-analysis of two studies evaluating CD4+ T cells [55,61,62] at POD1 (Fig. S3) also did not show a significantly different mean difference at POD1 […]" (page 11, lines 352-354)

Section 3.3.4: "The remaining four studies [48,52,56,61,62] did not report significant findings. [...] Meta-analysis including four studies [48,49,52,60] (Fig. S5) resulted in no significant differences between the groups at POD1 [...]." (page 11, line 360, lines 366-368)

As pointed out correctly by this reviewer, the references 61 and 62 (first submitted Version of the manuscript: references 60 and 61) both correspond to the same first author. This was also the case for Kvarnström et al. (references 53 and 54). As described in Figure 1 and section 3.1, these publications (records) were both based on an identical study population. This is why we always reference both publications but only speak of one study (we also quote both references in Table 1):

Section 3.1: "Of overall 16 records included, there were two cases of two records being based on the same study population (judged by population characteristics, authors, and publication date). Thus, 14 studies were included in this review of cellular immunity." (page 5, lines 213-216)

Reviewer 2 Report

This paper is very interesting, well written, methodologically well done, and contributes to the growing literature on lap vs open colorectal cancer surgery, this time in terms of cellular immunity.

Author Response

We very much thank the reviewer for this encouraging review.

Reviewer 3 Report

In this review the authors aim to focus on cellular immunity as well as the humoral aspects of postsurgical immunity in patients with colorectal cancer in the context of different surgical approaches (minimally invasive versus open surgery).

Materials and methods are unclear. I have the impression that the section on materials and methods does not clearly define the terms of analysis: which specific data are taken into account and then analyzed? Please try to describe better this section.

Furthermore, I have the following major concerns about the methodology:

1. Study selection and bias. The article mentions eligibility criteria stating that randomized controlled trials (RCTs) were systematically searched, but it doesn't provide details on the specific inclusion and exclusion criteria used to select the trials. It is crucial to assess the quality of all included studies. It is important to determine if any bias affecting a single trial may have influenced the overall results of the meta-analysis. In other words it is importan to know if the included trials had a low or high risk of biases or if there were concerns regarding the methodological quality of the studies considered

2. Sample size and heterogeneity of the studies included in this review. The Authors state that 14 trials with 974 participants were assessed, but they don’t provide information on the individual sample sizes and specific characteristics of each trial. Assessing the heterogeneity between the trials is crucial to determine if they were sufficiently similar and homogeneous to be combined in a meta-analysis.

3. Outcome Measures. The review mentions the analysis of various cell counts (NK cells, white blood cells, lymphocytes, CD4+ T cells, CD4+/CD8+ ratio), NK cell activity, and HLA-DR expression rates. It would be helpful to know if these outcome measures were predefined, consistent and all present across all included trials considered.

4. Generalizability. The review compared studies about laparoscopic surgery (LS) and open surgery (OS) in colorectal cancer patients of any stage. However, without additional information about different stage of disease, it is unclear if the findings are applicable to all patients with different stages of colorectal cancer. It is also unclear if certain stages (advanced tumour stage) were overrepresented in some of the included trials.

5. Publication bias: This review does not mention any assessment of publication bias. This refers to the tendency to publish studies with positive or statistically significant results that are more likely to be published than those with negative or non-significant results. How many studies have been published without significant results among those considered ?  It is critical in a meta-analysis to consider this potential bias, as it may impact the overall conclusions.

6. Need for Further Research: The review concludes that further research is required, suggesting that the existing evidence may not be conclusive. It would be helpful to know in detail the specific limitations of each included study and the areas where more research is needed. This will provide a clearer understanding of the influence and role of laparoscopic surgery on cellular immunity in colorectal cancer patient.

Round 2

Reviewer 3 Report

All questions have been satisfactorily answered by the authors.

I congratulate the authors for having answered all the questions posed by the reviewer with great accuracy and I consider this review to be an interesting study for future in-depth research on the impact of cellular immunity as well as humoral aspects of postsurgical immunity in patients with colorectal cancer when utilizing different minimally invasive techniques